# Impacts of Climate Change on Hydrological Regimes in the Congo River Basin

**Sara Karam** [1,*], **Baba-Serges Zango** [1], **Ousmane Seidou** [1,2,*], **Duminda Perera** [1,2,3], **Nidhi Nagabhatla** [4] and **Raphael M. Tshimanga** [5]

1    Faculty of Engineering, University of Ottawa, Ottawa, ON K1N 6N5, Canada
2    Institute for Water, Environment and Health, United Nations University, Hamilton, ON L8P 0A1, Canada
3    School of Earth, Environment and Society, McMaster University, Hamilton, ON L8S 4L7, Canada
4    Institute on Comparative Regional Integration Studies (UNU-CRIS), United Nations University, 8000 Bruges, Belgium
5    Congo Basin Water Resources Research Center—CRREBaC, Department of Natural Resources Management, University of Kinshasa, Kinshasa P.O. Box 117, Congo
*    Correspondence: skara010@uottawa.ca (S.K.); ousmane.seidou@uottawa.ca (O.S.)

**Abstract:** Surface water resources are essential for a wide range of human activities, such as municipal water supply, fishing, navigation, irrigation, and hydropower. Their regime is also linked to environmental sustainability, water-related risks, human health, and various ecosystem services. Global warming is expected to modify surface water availability, quality, and distribution and therefore affect water use productivity as well as the incidence of water-related risks. Thus, it is important for communities to plan and adapt to the potential impacts of climate change. The Congo River Basin, home to 75 million people, is subject to recurrent flood and drought events, which are expected to worsen as a result of climate change. This study aims to assess future modifications of the hydrological regime of the Congo River and the socio–economic impacts of these projected changes for three future periods: 2011–2041, 2041–2070, and 2071–2100. A Soil and Water Assessment Tool (SWAT) model of the Congo River Basin was developed, calibrated, and validated using daily rainfall observations combined with daily time series of precipitation, temperatures, relative humidity, solar radiation, and wind speed derived from the WFDEI (Watch Forced Era Interim) reanalysis data set. The outputs of ten Regional Climate Models (RCMs) from the Coordinated Downscaling Experiment (CORDEX-AFRICA) were statistically downscaled to obtain future climate time series, considering two Representative Concentration Pathways: RCP8.5 and RCP4.5. The calibrated model was used to assess changes in streamflow in all reaches of the Congo River. Results suggest relative changes ranging from −31.8% to +9.2% under RCP4.5 and from −42.5% to +55.5% under RCP 8.5. Larger relative changes occur in the most upstream reaches of the network. Results also point to an overall decrease in discharge in the center and southern parts of the basin and increases in the northwestern and southeastern parts of the basin under both emission scenarios, with RCP8.5 leading to the most severe changes. River discharge is likely to decrease significantly, with potential consequences for agriculture, hydropower production, and water availability for human and ecological systems.

**Keywords:** Congo River Basin; hydrological regime; water resources; drought; SWAT model; representative concentration pathway

## 1. Introduction

The Congo River Basin (Figure 1a) is located in the central region of Africa and spans over nine political boundaries. It is home to one of the largest watersheds in the world, with an estimated drainage area of 3.7 million square kilometers [1]. It is home to a growing population that currently stands at 75 million people. It is an important source of water, food, and transportation for the region, and it plays a vital role in the economies of the countries it passes through. The Congo basin is home to the second-largest rainforest in

the world, which plays a crucial role in the global carbon cycle. Like other rainforests, the distribution and makeup of the Congo basin's rainforest may be impacted by variations in local rainfall patterns due to global warming [2]. The Congo basin is a considerable source of major drought and storm events. Drought events in these regions are common and can affect thousands of people [3]. Evidence suggests that throughout the dry periods for the past 3000 years, the rainforest in the Congo basin has contracted and undergone significant changes in its composition [4,5]. It is becoming ever more important to study the climate changes occurring in the Congo basin in order to provide reliable information for planning related to mitigation and adaptation [6]. However, despite the importance of the Congo basin, it has not been adequately studied and has not received sufficient attention in research on hydrology and climate [7]. From 1960 to 2019, more than 11.5 million people in the countries of the Congo basin were affected by flooding. There were 3062 fatalities and severe economic losses that are estimated to be around $96 billion [8].

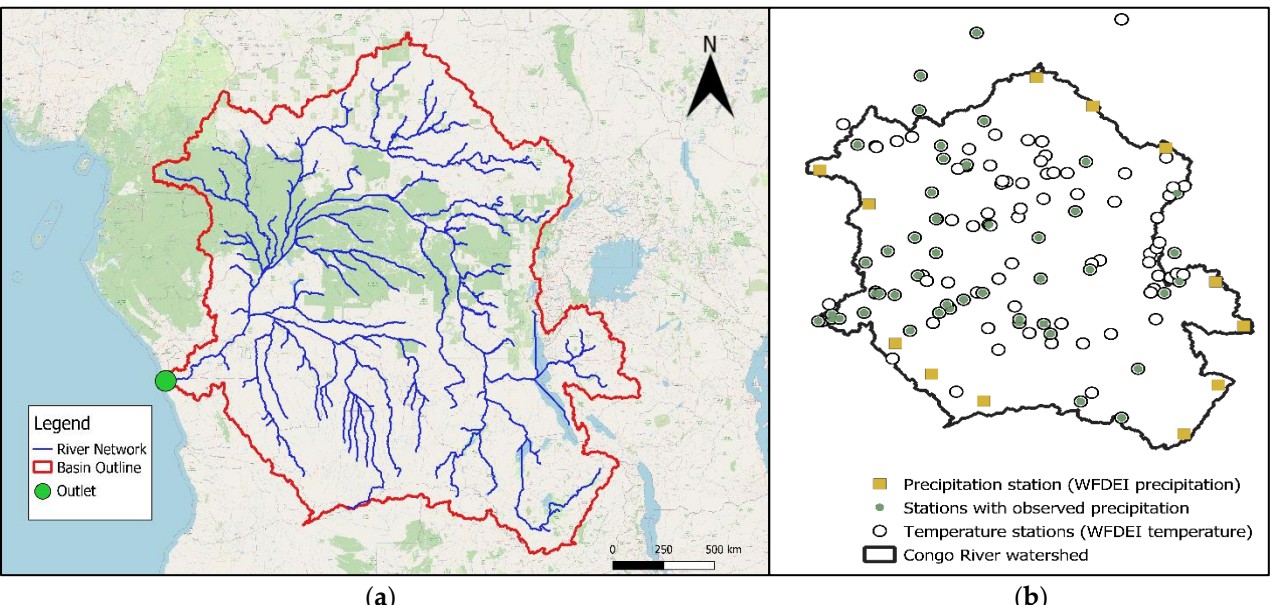

**Figure 1.** (**a**) Geographical location of the Congo River Basin with the river network and the outlet location and (**b**) location of observed rainfall stations and location of WFDEI precipitation and temperature stations in the Congo River Basin.

The general objective of this paper is to assess the impacts of climate change on the hydrological regime of the Congo River. Specific objectives include the development of a hydrological model of the Congo River Basin and the quantification of expected changes in average flow in three future periods (2011–2040, 2041–2070, and 2071–2100). The results will be spatially mapped in order to support climate change adaptation policies that will protect local infrastructure and ensure the sustainability of people's livelihoods.

Several studies have investigated the effects of climate change on hydroclimate variables in the study area. Tshimanga and Hughes (2014) [7] analyzed the effects of climate change on the hydrology of the Oubangui and Sangha watersheds, focusing on near-future runoff. The analysis was based on climate models (CMIP3 GCMs). Their research found that evapotranspiration is likely to significantly increase, and total runoff is expected to decline by 10%.

Sridhar et al. (2022) [9] evaluated the Congo River Basin's land use and land cover from 1992 to 2012, revealing a decrease in forests and native vegetation and a slight increase in urban and cropland areas. The study found that a framework combining the hydroclimate assessment of total water storage with hydrological SWAT models and remote sensing is feasible within the basin. Significant declines in forests and shrublands and increases in urban areas in the Oubangui and Middle Congo regions led to notable



changes in the water budget and an increase in streamflow. Variations in temperature and precipitation at the basin scale have had an impact on streamflow and exacerbated total water storage conditions. Projections for the future indicate that increased temperatures and variable precipitation will lead to sub-basin-scale differences, with overall increases in evapotranspiration and runoff and more frequent drought events in the basin.

Santos et al. (2022) [10] conducted a comprehensive comparison of various satellite-based precipitation products as inputs for a hydrological SWAT model in the Congo River Basin. The research found that products based on satellite-only sources tend to overestimate the peaks of the rainy season. However, satellite products that consider gauge calibration were found to have better agreements with one another. The overall precipitation patterns were found to have a significant impact on the model's performance, resulting in different values for streamflow and water balance components.

Kitambo et al. (2022) [11] used a large collection of in situ and satellite-derived data, including a long-term record of surface water height and surface water extent, to examine the surface hydrology and seasonal variability in the Congo River Basin. They found that the surface water height data from multiple satellite missions were consistent with in situ measurements and that the surface water extent data from various satellites accurately represented the hydrological patterns in the basin over a period of approximately 25 years. The data also revealed significant variability in the surface water height and extent in the Congo River, with annual amplitudes of over 5 m in certain northern subbasins and smaller variations in the main stream and Cuvette Centrale tributaries. Their results provide an understanding of the seasonal hydrological variability in the Congo River Basin.

Čerkasova et al. (2018) [12] created an application of the SWAT model that was used to develop a hydrology and water quality model for a vast watershed of the transboundary Vilija River in Europe. The primary aim was to investigate the impacts of climate change. It was found that although the RCP8.5 scenario represented the extreme range of projected changes, the hydrologic regime of the Vilija River is still expected to undergo significant changes in the RCP4.5 scenario. This is mainly due to the anticipated increase in precipitation, which is projected to be higher in the RCP4.5 scenario than in the RCP8.5 scenario.

Park et al. (2014) [13] conducted research to assess how climate change may affect different hydrological components of the Yongdam watershed (930 km$^2$), including evapotranspiration, surface runoff, lateral flow, return flow, and streamflow. The SWAT model was calibrated and validated using daily soil moisture and streamflow data. The results indicate that future climate change is expected to increase evapotranspiration, surface runoff, baseflow, and streamflow by 11.8%, 36.8%, 20.5%, and 29.2%, respectively. These findings suggest that future temperature and precipitation increases, as predicted by the RCP emission scenarios, will likely lead to an overall increase in hydrological patterns.

Anjum et al. (2019) [14] examined the potential effects of anticipated climate changes on the outflows of a humid subtropical basin located in the westerly dominated region of the Hindukush Mountains. Using six GCMs and two RCPs, 4.5 and 8.5, they downscaled precipitation and temperature projections and calibrated a SWAT model to find that climate changes could significantly impact the seasonality of river discharge.

The focus of these previous studies that consider the Congo basin are either seasonal studies on the variation of the hydrological regime or studies on the climatological properties surrounding the basin. In light of the significance of the study area and its vulnerabilities to hydrological variations, it is crucial to have a current understanding of food and drought patterns in the present and in the future. While extreme climatology of the Congo basin has been studied in the past, future hydrological regimes give us a more in-depth perception of the physical variations occurring within the basin, which gives a sharper idea of the potential vulnerability of the basin. By understanding hydrological conditions, it is possible to improve water resource management, reduce the risk of water-related disasters, and develop adaptation policies. This study aims to evaluate the potential changes in hydrological patterns in the Congo basin from the present until 2100.

## 2. Materials and Methods

### 2.1. Available Data

The Congo basin Water Resources Research Center (CRREBaC) facilitated monthly precipitation time series as well as maximum and minimum temperature time series. Daily rainfall observations ranged from the years 1961 to 2019 at the Kinshasa–Mbinza station; however, the authors were not able to obtain observations at other locations in the basin. Alternatively, precipitation, maximum and minimum temperatures, relative humidity, solar radiation, and daily wind speed time series were obtained from the Watch Forced Era Interim (WFDEI) reanalysis data set [15]. As measured station records are usually deemed to be more accurate than reanalysis data [16], there appeared to be a considerable amount of incorrect data in the observation records. Thus, any observed rainfall data in which the yearly totals are different by ten percent or more of that of the WFDEI data set were discarded. After this distinction, the observed rainfall stations lowered from 121 stations to 50 stations. The WFDEI was also used at additional stations to provide appropriate spatial coverage of the study area for a total of 87 precipitation stations. Given that the WFDEI database derives a temperature estimate that provides higher accuracy than precipitation estimates, the authors chose to utilize the WFDEI time series for 128 stations. Similarly, WFDEI was used for relative humidity, solar radiation, and wind speed at 82 locations. The geographical location of the Congo basin and the outlet location (gauge used for calibration and validation) can be observed in Figure 1a, and the observation stations for precipitation and temperature are shown in Figure 1b.

Ten climate change experiments were conducted by the Coordinated Downscaling Experiment (CORDEX) group (https://cordex.org/domains/region-5-africa/, accessed on 31 January 2022) [17] to gather information on projected precipitation and maximum and minimum temperatures from 1950 to 2100. The trials entailed utilizing GCMs to drive the RCMs under two circumstances, namely RCP4.5 and RCP8.5.

To generate a statistical distribution of a climate parameter that closely matches the observed parameter in the historical period, the quantile–quantile (Q–Q) transformation is used. As per Hakala et al. (2018) [18], this method outperforms other approaches, such as the delta-change method, local intensity scaling, and power transformation in reducing biases in GCM–RCM precipitation. The detailed procedure conducted in this paper for bias correction is described by Karam et al. (2022) [19]. The details of the institutions and the driving GCM and RCM are presented in Table 1.

**Table 1.** Characteristics of the GCM/RCM model combinations used in the study.

| No. | Driving GCM | GCM Institute | RCM | RCM Institute |
|---|---|---|---|---|
| 1 | Canadian Earth System Model Version 2 | Canadian Centre for Climate Modeling and Analysis | Canadian Regional Climate Model 4 | Canadian Centre for Climate Modeling and Analysis |
| 2 | Canadian Earth System Model Version 2 | Canadian Centre for Climate Modeling and Analysis | Rossby Centre regional atmospheric model, version 4 | Swedish Meteorological and Hydrological Institute |
| 3 | Centre National de Recherches Météorologiques, Climate Model 5 | National Centre for Meteorological Research | Rossby Centre regional atmospheric model, version 4 | Swedish Meteorological and Hydrological Institute |
| 4 | Queensland Climate Change Centre of Excellence (QCCCE) and Commonwealth Scientific and Industrial Research Organization (CSIRO) | The Commonwealth Scientific and Industrial Research Organization | Rossby Centre regional atmospheric model, version 4 | Swedish Meteorological and Hydrological Institute |

| No. | Driving GCM | GCM Institute | RCM | RCM Institute |
|-----|-------------|---------------|-----|---------------|
| 5 | European community Earth-System Model | Irish Centre for High-End Computing | Rossby Centre regional atmospheric model, version 4 | Swedish Meteorological and Hydrological Institute |
| 6 | Institut Pierre Laplace Climate Model version 5A | Institute Pierre Simon Laplace | Rossby Centre regional atmospheric model, version 4 | Swedish Meteorological and Hydrological Institute |
| 7 | Model for Interdisciplinary Research on Climate, version 5 | Center for Climate System Research/National Institute for Environmental Studies/Frontier Research Center for Global Chance, Japan Agency for Marine-Earth Science and Technology | Rossby Centre regional atmospheric model, version 4 | Swedish Meteorological and Hydrological Institute |
| 8 | Max Plank Institute Earth System Model | Max Planck Institute for Meteorology | Rossby Centre regional atmospheric model, version 4 | Swedish Meteorological and Hydrological Institute |
| 9 | Norwegian Earth System Model | Norwegian Climate Centre | Rossby Centre regional atmospheric model, version 4 | Swedish Meteorological and Hydrological Institute |
| 10 | Geophysical Fluid Research Laboratory Earth System Model | National Oceanic and Atmospheric Administration-Geophysical Fluid Dynamics Laboratory | Rossby Centre regional atmospheric model, version 4 | Swedish Meteorological and Hydrological Institute |

### 2.2. Model Setup

The SWAT (Soil and Water Assessment Tool) is a hydrological and water-quality model developed by the Agricultural Research Service (ARS) of the United States Department of Agriculture (USDA). It was developed by Arnold et al. in 1998 [20] and is a physically based, semi-distributed continuous model used to simulate the impact of land management practices on water, sediment, and agricultural chemical yields in large, complex watersheds.

ArcSWAT was developed as an extension to the ArcGIS software version 10.7, and was used in conjunction with SWAT as a graphical user interface. This tool combines spatial and climatic inputs to set up and run the SWAT model and to visualize and analyze the results of the models within the ArcGIS platform [21]. ArcSWAT comes with a global database of default parameters which allows users to quickly develop a functional model of any area. ArcSWAT discretizes the watershed into sub-watersheds and Hydrological Response Units (HRUs) based on user-specified location of outlets and the HRU definition method. An HRU is a unique combination of topographical, land-use management, and soil characteristics [22].

The hydrological simulations conducted in the SWAT use the water balance equation which is utilized at the HRU scale for each time step:

$$SW_t = SW_0 + \sum_{i=1}^{t} \left( R_{day} - Q_{surf} - E_a - w_{seep} - Q_{gw} \right) \tag{1}$$

where $SW$ is the soil water content at time 0 and at a time step, $t$. $R_{day}$ is the precipitation amount, $Q_{surf}$ is the surface runoff, $E_a$ is the evapotranspiration, $w_{seep}$ is the water entering the vadose zone from the soil profile, and $Q_{gw}$ is the return flow on a specific day, $I$, in

units of mm. Rainfall is then subdivided into infiltration and surface runoff using the curve number (CN) method [23]:

$$Q_{surf} = \frac{(P - 0.2)^2}{(P - 0.8S)} \tag{2}$$

where the surface runoff ($Q_{surf}$) is derived as a function of the total precipitation ($P$) in mm and the soil moisture content ($S$), a no-dimension parameter expressed in terms of a CN ranging from 0 to 100. The constants 0.2 and 0.8 are coefficients that are used to adjust the relationship between the surface runoff and the other variables. The surface runoff increases as the total precipitation increases. This is because more precipitation leads to more runoff. The surface runoff decreases as the soil moisture content increases. This is because a higher soil moisture content means that more of the precipitation will be absorbed by the soil and less will run off.

A total of 223 sub-watersheds and twelve HRUs were delineated (Figure 2), and land use characteristics of the Congo River Basin are depicted in Table 2. The remainder of the model development consists of setting up and calibrating the model, validating the model's predictions, and analyzing and interpreting the results to inform land use and management decisions.

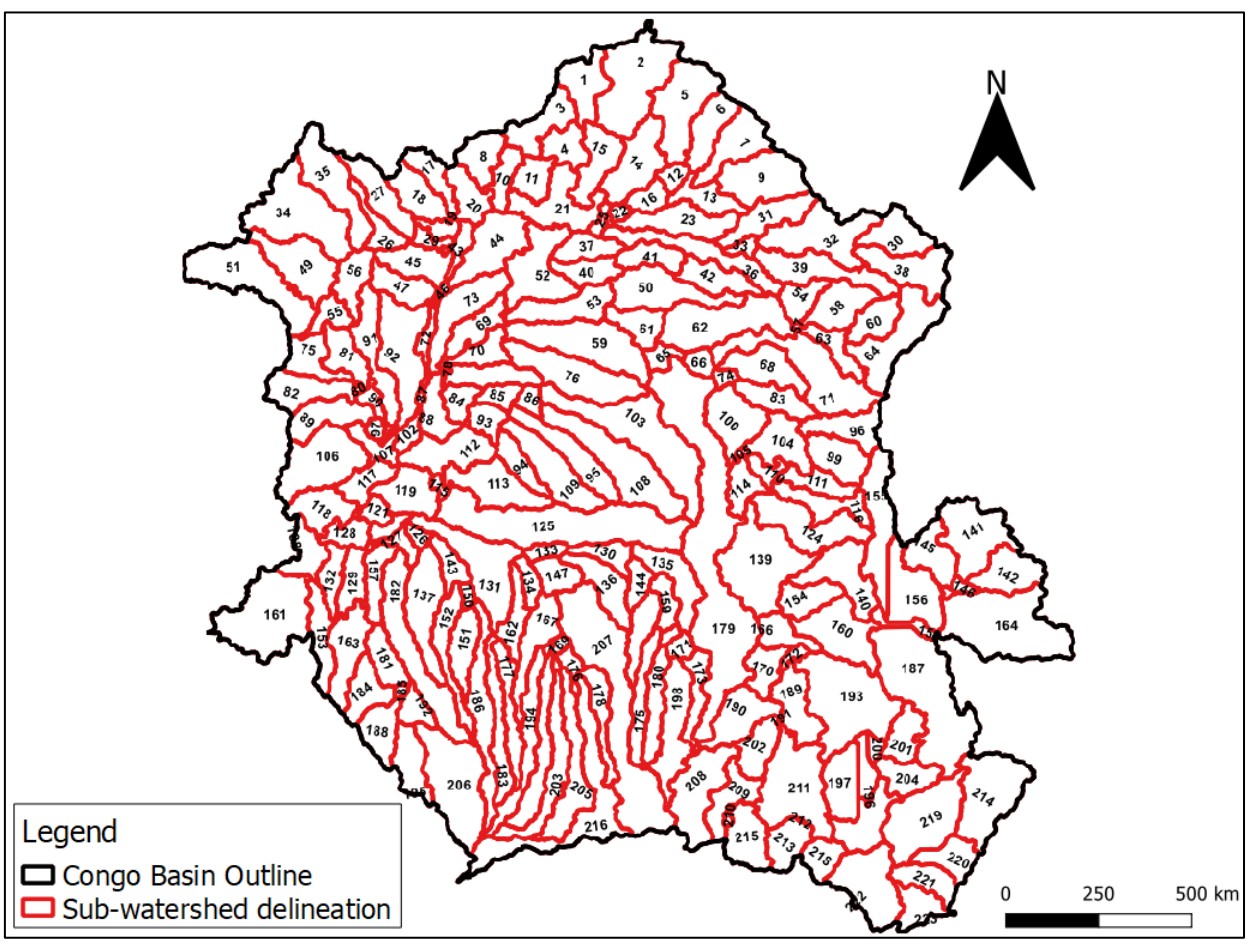

**Figure 2.** Sub-watersheds in the Congo basin model.

**Table 2.** Land-use characteristics of the Congo River Basin [22].

| Land Use/Cover Type | SWAT Code | Area (%) |
|---|---|---|
| 1. Water | WATR | 2.70 |
| 2. Residential area (urban) | URMD | 0.02 |
| 3. Dryland cropland and pasture | CRDY | 3.87 |
| 4. Mosaic cropland/grassland | CRGR | 0.12 |
| 5. Mosaic cropland/woodland | CRWO | 7.15 |
| 6. Grassland | GRAS | 0.19 |
| 7. Shrubland | SHRB | 0.31 |
| 8. Savanna | SAVA | 28.27 |
| 9. Deciduous broad-leaf forest | FODB | 13.70 |
| 10. Evergreen broad-leaf forest | FOEB | 42.89 |
| 11. Mixed forest | FOMI | 0.18 |
| 12. Barren or sparsely vegetated | BSVG | 0.58 |

*2.3. Calibration and Validation*

Calibration and validation are important steps in the model development process, as they help to ensure that the model is accurately representing the processes that are being simulated. The calibration process involves adjusting the model parameters and input data to improve the accuracy of the model predictions. The validation process involves evaluating the performance of the calibrated model using additional data that was not used in the calibration process. This helps to determine the accuracy and reliability of the model under different conditions and to identify any potential biases or errors in the model.

2.3.1. Performance Measures

The performance of the calibration and validation is evaluated using statistical and graphical methods. This study uses the Nash–Sutcliffe efficiency coefficient (NS) [24] and the percentage bias (PBIAS) [25] as statistical measures of the accuracy and performance of the model. The NS coefficient is a dimensionless measure that ranges from $-\infty$ to 1, with values closer to 1 indicating a better fit between the observed and simulated values. It is given by the following equation:

$$NS = 1 - \frac{\sum_{i=1}^{n}(Q_m - Q_s)_i^2}{\sum_{i=1}^{n}\left(Q_{m,i} - \overline{Q}_m\right)^2} \tag{3}$$

where $Q$ is the discharge, $m$ is the measured data, $s$ is the simulated data, $i$ is the time step, and $n$ is the total number of periods.

The PBIAS is another measure of model accuracy that is calculated by comparing the mean of the observed values to the mean of the simulated values and expressing the difference as a percentage of the observed mean. A PBIAS value of 0 indicates that the model is unbiased, while positive and negative values indicate a tendency for the model to underestimate or overestimate the observed values, respectively.

$$PBIAS = 100 \times \frac{\sum_{i=1}^{n}(Q_m - Q_s)_i}{\sum_{i=1}^{n}Q_{m,i}} \tag{4}$$

The statistical measures obtained in this paper will be compared to the model performance criteria illustrated in Table 3 [26].

**Table 3.** Model performance criteria [26].

|  | **Satisfactory** | **Good** | **Very Good** |
|---|---|---|---|
| NS | 0.5–0.7 | 0.7–0.8 | 0.8–1.0 |
| PBIAS (%) | 15–25 | 10–15 | Less than 10% |

2.3.2. Sensitivity Analysis

The calibration and the sensitivity analysis were conducted using the SUFI2 (Sequential Uncertainty Fitting) algorithm produced in the SWATCUP program [27]. The sensitivity analysis is used to evaluate the sensitivity of the model's output to changes in different input parameters. It is computed by altering one parameter while all other parameters remain unchanged. The effectiveness of the changed parameter is then evaluated with t-stat and *p*-value, which are determined with a multiple regression analysis using the following equation [27]:

$$g = \alpha + \sum_{i=1}^{m} \beta_i b_i \tag{5}$$

where *g* is the objective function used to determine the model calibration effectiveness, *b* is the parameter in question, *α* is equal to the regression constant, *β* is the technical coefficient attached to the variable *b*, and m corresponds to the number of parameters.

The *p*-value is a statistical measure that represents the probability of obtaining a test statistic at least as extreme as the one observed, assuming that the null hypothesis is true. The t-statistic is used to determine whether the mean of a population is significantly different from a hypothesized value. If the absolute value of t-stat is relatively high and the *p*-value is relatively small, the parameter is considered to have a high sensitivity [27]. Typically, a *p*-value lower than 0.05 indicates that the parameter is significantly sensitive [28].

Given the large number of parameters of the SWAT model, only a small subset is calibrated. That subset was selected after a sensitivity analysis, which consists of slightly modifying the parameter values and seeing their impact. Choosing sensitivity parameters for calibration requires a combination of expert knowledge, sensitivity analysis, and trial and error. Initially, the most important input parameters must be identified; then, a reasonable range of values must be chosen. This range should be based on the published literature or previous modeling studies. If the model performance is not satisfactory, the sensitivity parameters should be adjusted and the calibration process should be repeated.

Twelve parameters were chosen for the calibration with the help of several recommendations by authors who conducted similar studies [22,29–32]. The parameters included in the sensitivity analysis are shown in Table 4.

**Table 4.** Parameters used for calibration/validation.

| Parameter Name | Parameter Description | Variation Range |
|---|---|---|
| r_HRU_SLP.hru | Average slope steepness (m/m) | 0.1–0.5 |
| r_ESCO.bsn | Soil evaporation compensation factor | 0.1–0.5 |
| r_EPCO.bsn | Plant uptake compensation factor | 0.1–0.4 |
| r_REVAPMN.gw | Threshold depth of water in the shallow aquifer for 'revap' to occur (mm) | 0.1–1.0 |
| r_GWQMN.gw | Threshold water depth in the shallow aquifer for flow | 0–0.5 |
| r_GW_REVAP.gw | Groundwater 'revap' coefficient | −0.1–0.1 |
| r_SOL_AWC.sol | Available water capacity | −0.5–0.1 |

**Table 4.** *Cont.*

| Parameter Name | Parameter Description | Variation Range |
|:---:|:---:|:---:|
| r_ESCO.hru | Soil evaporation compensation factor | 0–0.3 |
| r_ALPHA_BF.gw | Baseflow alpha factor | 0.01–1 |
| r_CN2.mgt | Initial SCS CNII value | −0.2–0.5 |
| r_RCHRG_DP.gw | Deep aquifer percolation fraction | 0–10 |
| r_SOL_BD.sol | Moist bulk density ($Mg/m^3$ or $g/cm^3$) | 0.01–2.0 |

## 3. Results and Discussion

### 3.1. Sensitivity Analysis

The results of the global sensitivity analysis are shown in Table 5. The parameter with the most influence on the performance of the model is the deep aquifer percolation fraction (RCHRG_DP) with a t-stat and *p*-value of −0.09 and 0.92, respectively, whereas the least sensitive parameter is the average slope steepness (HRU_SLP) with a t-stat and a *p*-value of 0 and 0.99, respectively.

**Table 5.** Sensitivity analysis result.

| Parameter Name | t-Stat | *p*-Value | Rank |
|:---:|:---:|:---:|:---:|
| r_RCHRG_DP.gw | −0.09 | 0.92 | 1 (most sensitive) |
| r_SOL_AWC.sol | −0.06 | 0.95 | 2 |
| r_SOL_BD.sol | −0.05 | 0.96 | 3 |
| r_CN2.mgt | −0.04 | 0.97 | 4 |
| r_REVAPMN.gw | −0.02 | 0.97 | 5 |
| r_GW_REVAP.gw | 0.03 | 0.97 | 6 |
| r_GWQMN.gw | −0.02 | 0.98 | 7 |
| r_EPCO.bsn | 0.01 | 0.99 | 8 |
| r_ESCO.hru | 0.01 | 0.99 | 9 |
| r_ALPHA_BF.gw | 0.01 | 0.99 | 10 |
| r_ESCO.bsn | 0.01 | 0.99 | 11 |
| r_HRU_SLP.hru | 0.00 | 0.99 | 12 (least sensitive) |

### 3.2. Calibration and Validation Performance

The calibration was performed for twelve parameters identified in the previous section. The outputs of the calibrated and validated model are shown in Figure 3, while the optimal fitted values for each parameter are shown in Table 6. The model demonstrates good performance for the discharge at the watershed outlet in calibration. According to Moriasi et al. (2007) [26], the NS coefficient is in the satisfactory range (0.5 < NS = 0.64 < 0.7), and the PBIAS = 1.2% is also very good (near 0). In the validation stage, an excellent performance was achieved with an NS equivalent to 0.88 and a PBIAS of −3.6%. However, the graphical comparison between the observed and simulated values shows that the model often tends to underestimate the peaks, which has proven to be a common problem in rainfall-runoff modeling [29,33]. This discrepancy can be due to the quality of climate data input and the spatial variation of the study area. In fact, spatial repartition of rain gauges and data scarcity of such a large catchment may lead to extreme precipitation events being overlooked, notably in large areas with a single rain gauge, where intense precipitation events would not be detected.

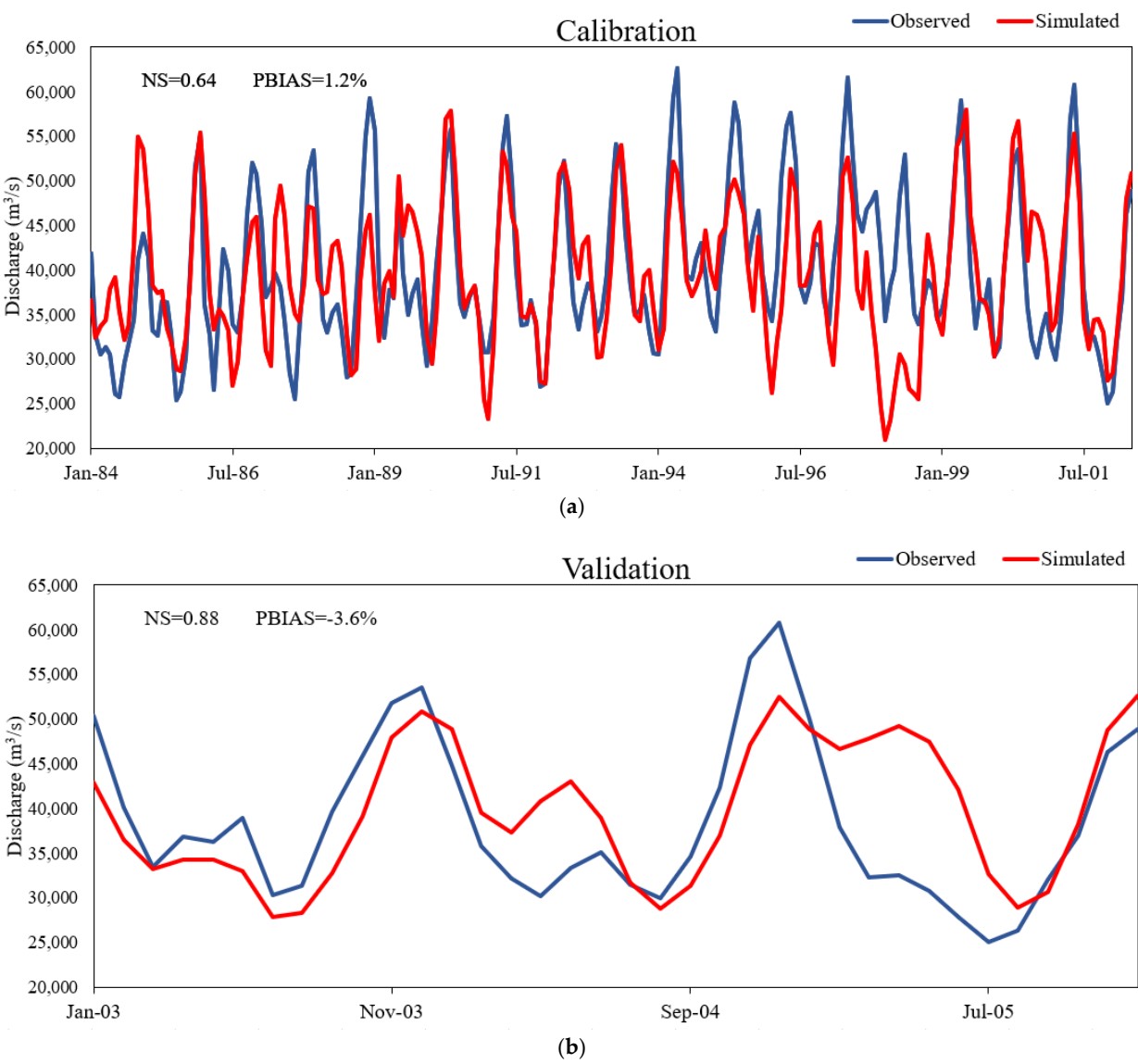

**Figure 3.** (**a**) Calibration and (**b**) validation performance at the defined Congo River Basin outlet.

**Table 6.** Parameters used for calibration/validation.

| Parameters | Optimal Fitted Values |
|:---:|:---:|
| r_HRU_SLP.hru | 0.447 |
| r_ESCO.bsn | 0.345 |
| r_EPCO.bsn | 0.249 |
| r_REVAPMN.gw | 0.917 |
| r_GWQMN.gw | 0.194 |
| r_GW_REVAP.gw | −0.068 |
| r_SOL_AWC.sol | −0.362 |
| r_ESCO.hru | 0.140 |
| r_ALPHA_BF.gw | 0.268 |
| r_CN2.mgt | −0.298 |
| r_RCHRG_DP.gw | 7.979 |
| r_SOL_BD.sol | 1.360 |

### 3.3. Projections for River Discharge

Figure 4 shows the average annual river discharge in cubic meters per second (cms) in the study area for a reference period of 1976 to 2005. The river discharge in this period ranges from 97 to 67,981 cms, with an average discharge value of 4713 cms. The highest discharge value can be observed at the outlet, while the southern part of the basin is the largest section in the basin with the least amount of discharge. In fact, minimal discharge can be observed in the extremities of the entire basin, except for near the outlet.

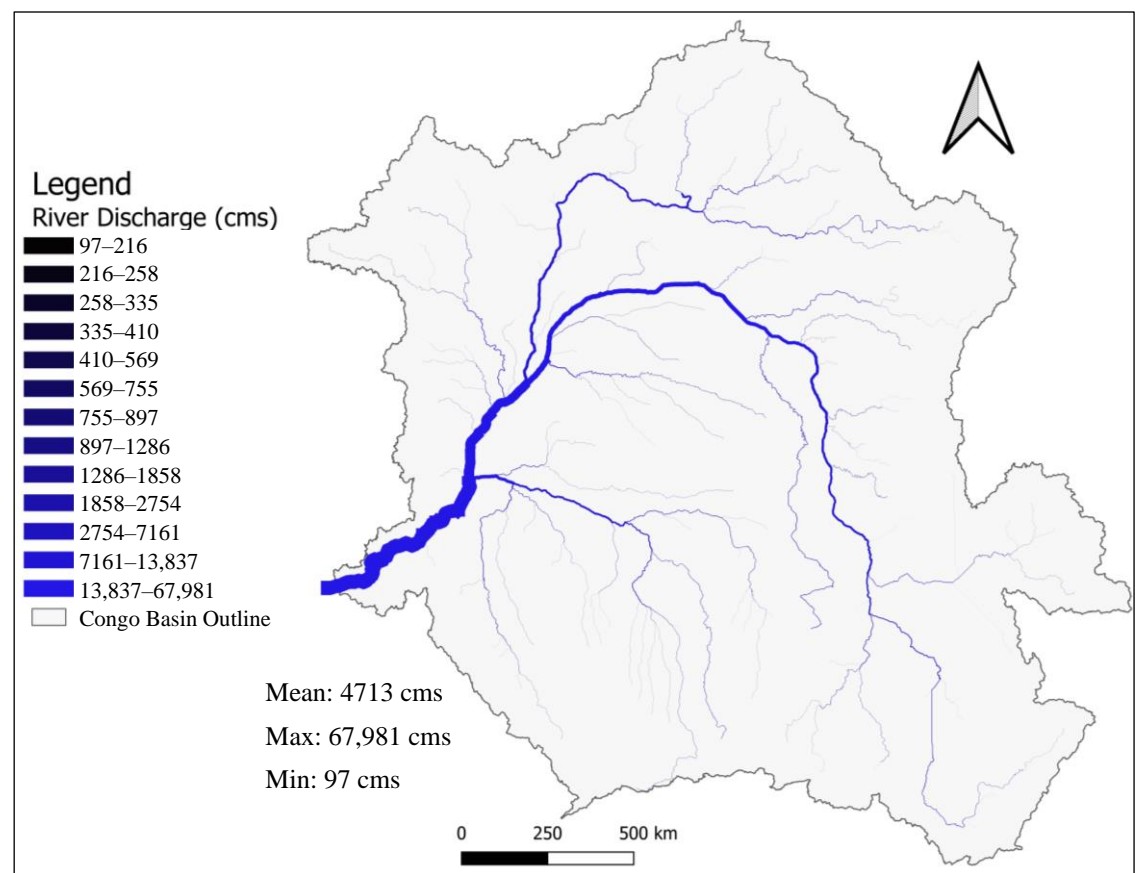

**Figure 4.** Average annual Congo River Basin discharge (cms) during reference period (1976–2005).

Figure 5 shows the average annual Congo River Basin discharge (cms) for RCP 4.5 for three future periods (2011–2040); (2041–2070); (2071–2100). These projections do not show any extreme variations between periods; however, a slight decrease in average annual discharge can be observed. The average discharge for period 1, period 2, and period 3 are 4597, 4417, and 4423 cms, respectively. This also predicts a decrease in average annual flow since the reference period (4713 cms). The maximum discharge located at the outlet of the basin for period 1, period 2, and period 3 are 66,382, 63,827, and 63,745 cms, respectively. Throughout all periods, the highest discharge value can be observed in the western stream leading towards the outlet. Similar to the reference period, the southern part of the basin is the largest section in the basin with the least amount of discharge, and minimal discharge can be observed in the extremities of the entire basin, except for near the outlet.

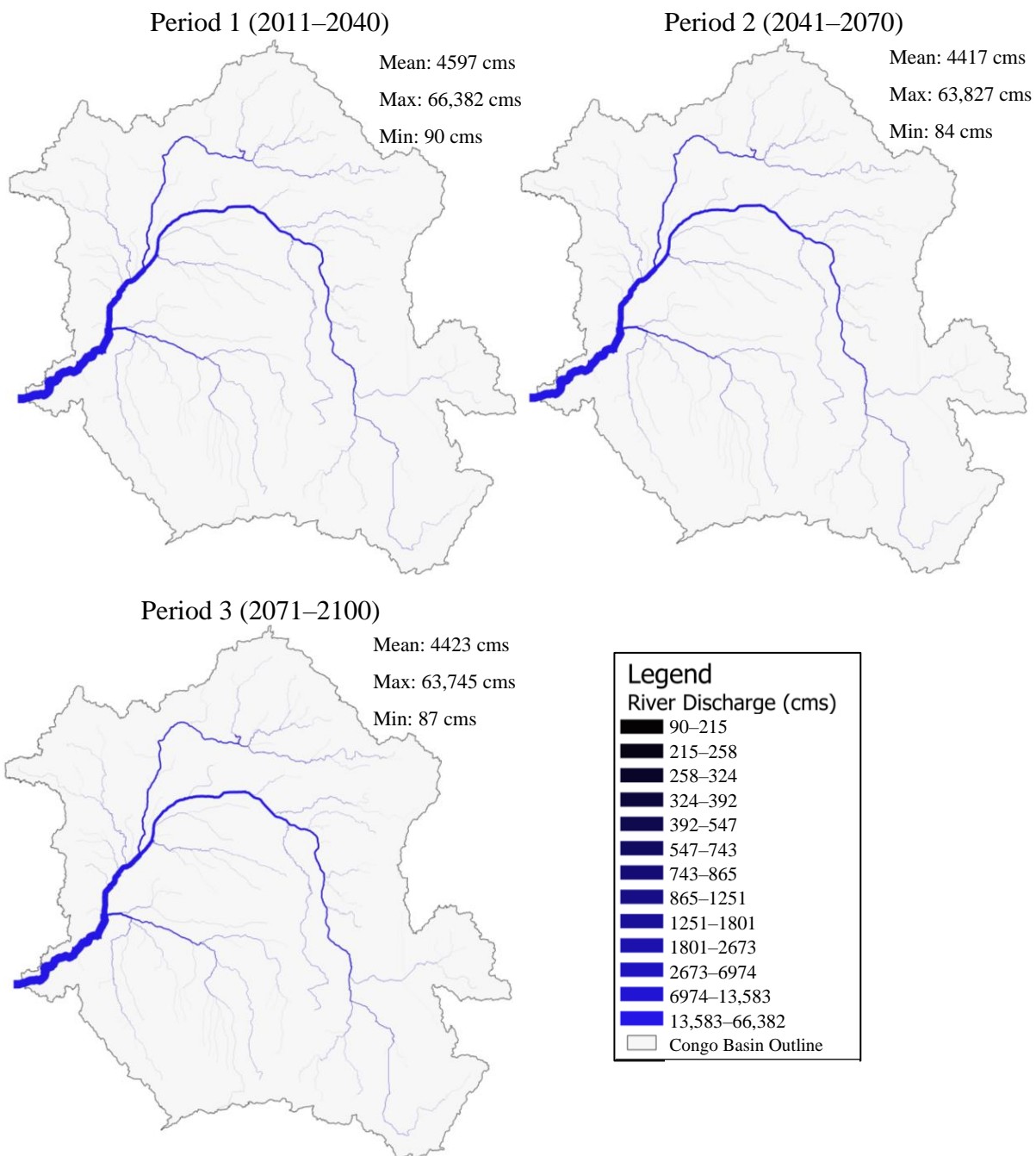

**Figure 5.** Average annual Congo River Basin discharge (cms) for RCP 4.5 for three future periods (2011–2040); (2041–2070); (2071–2100).

Figure 6 shows the percent change in average annual discharge for RCP 4.5 for three future periods (2011–2040); (2041–2070); (2071–2100). The average percent change of discharge for period 1, period 2, and period 3 are −1.8%, −6.1%, and −6.0%, respectively. The decrease in river discharge can be observed, especially in the center and southwestern sections of the basin, while the extremities of the basin show the most increase. The maximum and minimum changes occur in period 3 and period 2, with 9.2% change and −31.8% change, respectfully. Similar to what was represented in Figure 5, an overall decrease of the average discharge can be observed.

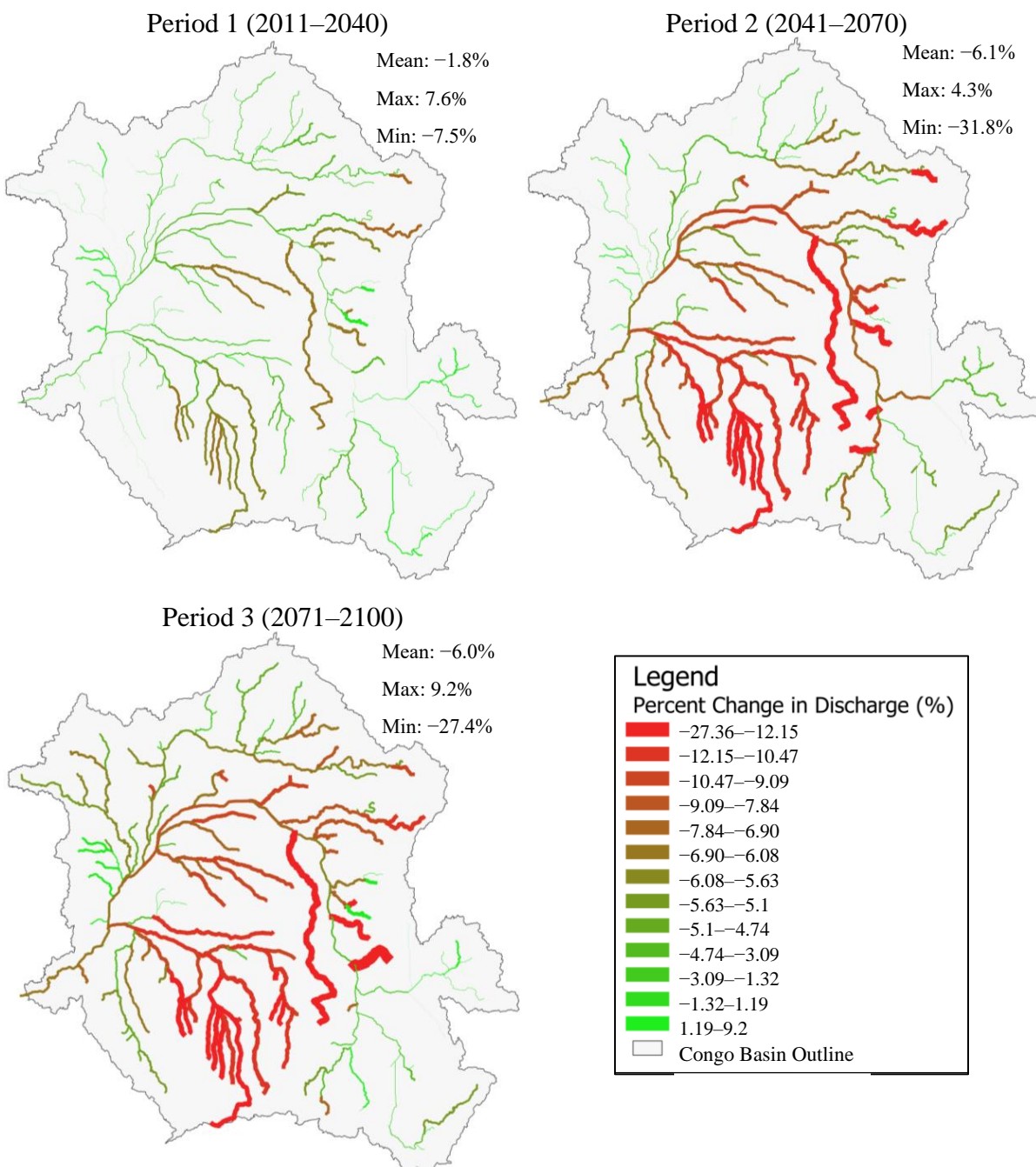

**Figure 6.** Percent change of average annual Congo River Basin discharge (%) for RCP 4.5 for three future periods (2011–2040); (2041–2070); (2071–2100).

Figure 7 shows the average annual discharge (cms) in the Basin for RCP 8.5 for three future periods (2011–2040); (2041–2070); (2071–2100). Similar to Figure 5, these projections do not show any extreme variations between periods; however, there is a fluctuation in average annual discharge where average flow in period 2 decreases from period 1, and then there is a significant increase in period 3 above that of the first period. The maximum discharge located at the outlet of the basin, for period 1, period 2, and period 3 are 65,337, 65,259, and 67,870 cms, respectively. When looking at the mean value of discharge in all three periods, a decrease in average annual flow since the reference period (4713 cms) can be observed. Similar to the projections using RCP4.5, RCP8.5 predicts throughout all periods that the highest discharge value is located near the outlet. The southern part of the

basin is the largest section in the basin with the least amount of discharge, and minimal discharge can be observed in the extremities of the entire basin, except for near the outlet.

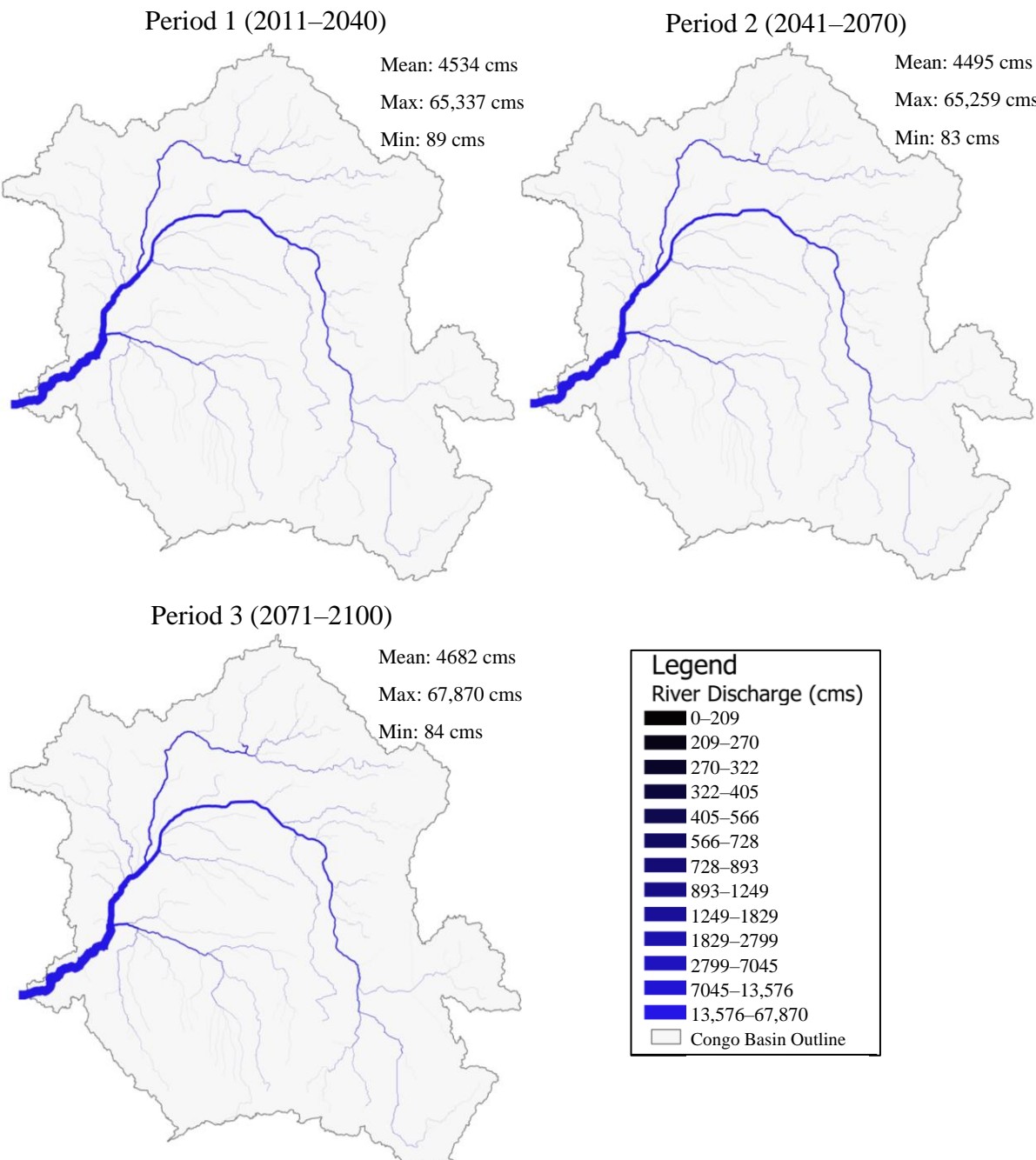

**Figure 7.** Average annual Congo River Basin discharge (cms) for RCP 8.5 for three future periods (2011–2040); (2041–2070); (2071–2100).

Figure 8 shows the percent change in average annual discharge for RCP 8.5 for three future periods (2011–2040); (2041–2070); (2071–2100). The average percent change of discharge fluctuates between periods. The change increases in the second period then descends nearly to zero in the final period. The average percent change of discharge for period 1, period 2, and period 3 are −3.1%, −3.9%, and 0.25%, respectively. The decrease in discharge can be observed, especially in the middle and the southwest areas of the watershed, while the northwestern section shows a significant increase. There is also

a stream in the southeast that shows a maximum increase of flow. The maximum and minimum changes occur in period 3 with values of 55.5% and −42.5%, respectfully.

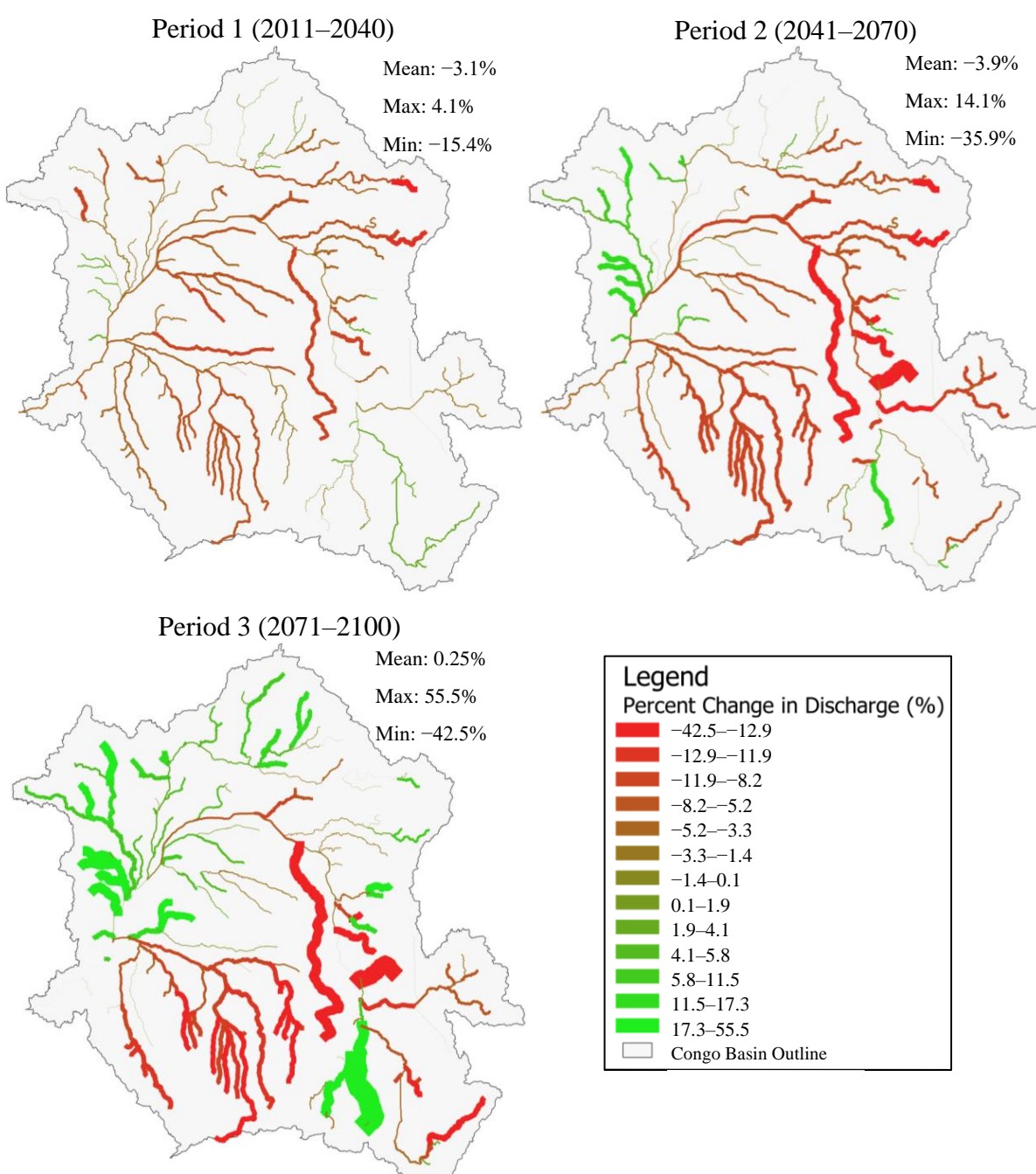

**Figure 8.** Percent Change of average annual Congo River Basin discharge (%) for RCP 8.5 for three future periods (2011–2040); (2041–2070); (2071–2100).

## 4. Discussion

### 4.1. Correlation with Climatological and Hydrological Studies

The results from this paper are consistent with a previous study by the authors where the future projections of climate extremes for the Congo River Basin were analyzed. Karam et al. (2022) [19] projected several climate indices, including yearly precipitation (PCPTOT), the number of days in a year where precipitation was above 20 mm (PCP20), the Standardized Precipitation Index (SPI), as well as the Standardized Precipitation-Evaporation Index (SPEI), over the entire Congo basin [19]. In their paper, the authors

evaluated the results of 11 regional climate models that utilized RCP4.5 and RCP8.5. The authors also examined the abovementioned climate indices for similar future periods and observed an increase in PCPTOT and PCP20 in the western and northern areas of the basin. The SPEI index demonstrated that there will be an increase in drought in most of the watershed area, and the authors suggested that this is due to the increase in evapotranspiration, which will outweigh the rise in precipitation due to a consistent annual increase of temperatures. Figure 9 below is an adaptation of the spatial variation of multi-model average SPEI results published by Karam et al. (2022) [19] showing that evaporation will offset the increase of PCPTOT in future periods under RCP4.5 and RCP8.5. The spatial average of SPEI in both RCP scenarios are exponentially decreasing over the basin area; in RCP4.5, for periods 1, 2, and 3, the SPEI average is 0.12, 0.07, and −0.24, respectively, and in RCP8.5, for periods 1, 2, and 3, the SPEI average is −0.21, −0.51, and −0.62, respectively.

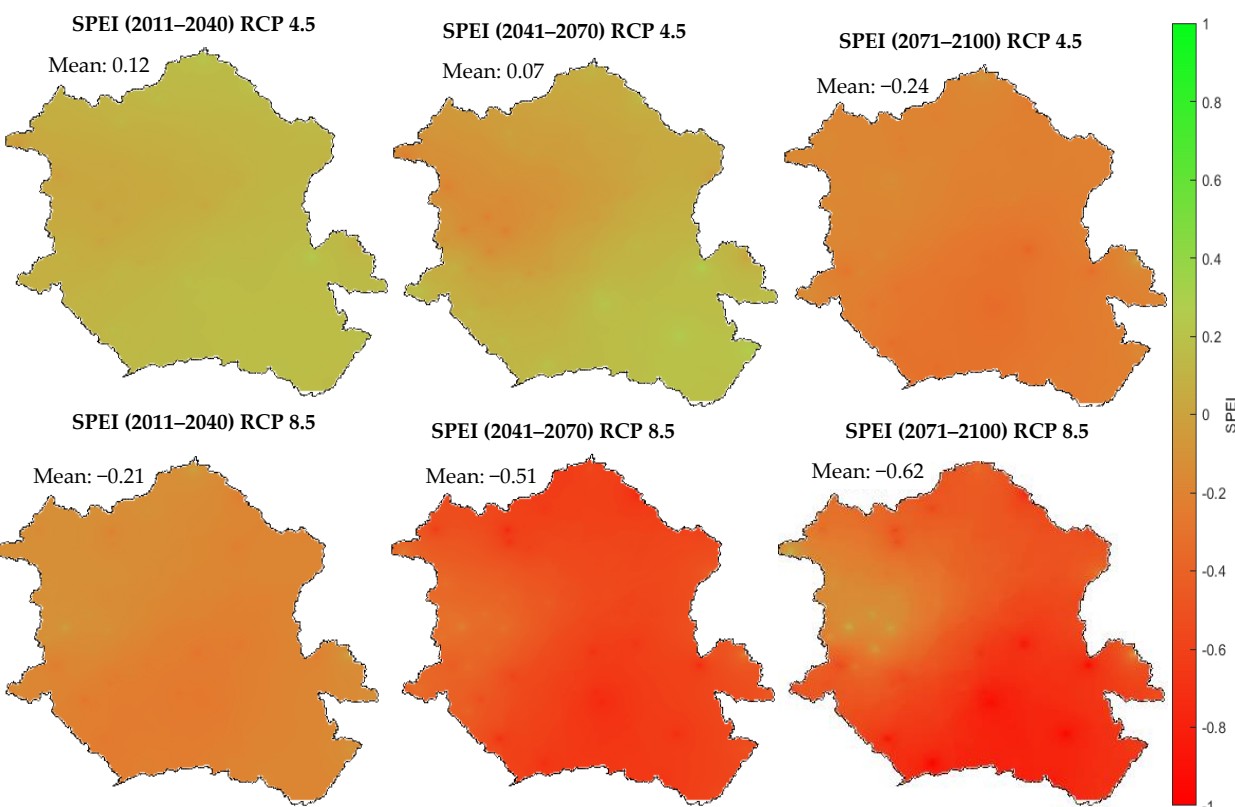

**Figure 9.** Spatial variation of multi-model average SPEI for three future periods (2011–2040); (2041–2070); (2071–2100).

The results from the climatic study conducted by Karam et al. (2022) [19] correlate well with the results obtained from the hydrological study conducted in this paper. The rise in temperatures and higher drought regimes may lead to an overall decrease in discharge in the middle and southern regions of the study area. Yet, high-intensity rainfalls and increased annual precipitation may lead to increased discharge, especially in the northern and western extremities of the basin. This investigation illustrates a basin that is expected to experience an escalation in both drought and flood occurrences in the near and late future.

Moreover, a hydrological study of the Congo basin conducted by Tshimanga et al. (2012) [34] downscaled GCM data to assess the impacts of climate change on the hydrology of the basin. The authors found that changes in river discharge would vary spatially and estimated a decrease in runoff for near-future projections. Major changes in the basin relate to increase in evapotranspiration due to higher air temperature. The results correlate well

with the findings of this study, which emphasize the urgent need for effective adaptation strategies to mitigate the impacts of climate change on the hydrology of the Congo basin.

Aloysius and Saiers (2017) [35] found that precipitation is likely to become more variable, and discharge is likely to increase in the short term (between 2016 and 2035) and vary considerably in the long term (between 2046 and 2065). Compatible with the findings of this study, Aloysius and Saiers (2017) [35] produced valuable insight demonstrating that changes in river discharge could have significant impacts on the water resources, ecosystems, and livelihoods of the region.

In the case where decision-makers have to use this information, it is important that the spatiotemporal variability of the impacts is considered appropriately for adaption planning and resilience building. For example, in sub-regions of the basin where projections point to an increase in drying conditions or droughts, priority and emphasis should be given to water storage approaches, incentives for water conservation practices for famers, and the promotion of water efficiency focused irrigation or water use methods. It is also important that water security and food security institutions and policies take note of these projects at the sub-national level and design interventions and 'fit to purpose' solutions for the local context.

### 4.2. Potential Impacts of Projected Changes

The hydrological regime of the Congo basin influences the lives of 75 million habitants. Yet, insufficient observational data has contributed to the basin being inadequately studied [6].

An overall decrease in average annual discharge and increased fluctuating peak discharge are expected to have significant socio–economic impacts on the Congo River Basin. This may impact agricultural production, water resources, health, and ecosystems. The projected decrease in average flow rates will likely affect agricultural production in the region, which could have negative impacts on food security and the livelihoods of farmers. It could also impact the availability of clean water and the ability of communities to rely on the river for irrigation and/or transportation. The hydrological fluctuations could also lead to the loss of biodiversity and the degradation of ecosystems, which could have negative impacts on the people who rely on these ecosystems for their livelihoods.

The socio–economic impacts of climate change are likely to be significant and far-reaching in the Congo River basin, and it will be important for communities and governments to work together to adapt to and mitigate these impacts.

## 5. Conclusions

The future hydrological conditions of the Congo basin were evaluated for three time periods (2011–2041, 2041–2070, and 2071–2100). A SWAT model was calibrated and validated and used to assess the impacts of climate change on water quantity. The model was run using daily rainfall observations combined with daily data on precipitation, temperatures, relative humidity, solar radiation, and wind speed from the WFDEI dataset. Future climate data was obtained by statistically downscaling the output of ten Regional Climate Models under RCP4.5 and RCP8.5.

To summarize, the projections suggest four key highlights: first, the rise in temperatures can lead to high incidents of drought regimes in the basin and an overall decrease in discharge in the center and southwestern parts of the basin; second, the high-intensity rainfalls and increase in annual precipitation may lead to increased discharge, especially in the northern and northwestern extremities of the basin; third, the maximum and minimum changes occur in period 3 under RCP 8.5, with values of 55.5% and −42.5%, respectfully, indicating that the high emissions scenario displays the most significant discharge variations throughout the basin in the next few decades; and lastly, the projections do not show any extreme variations between periods—however a slight decrease in average annual discharge is observed. The average discharge under RCP4.5 for period 1, period 2, and period 3 are 4597, 4417, and 4423 cms, respectively. The average discharge under RCP8.5

for period 1, period 2, and period 3 are 4534, 4495, and 4682 cms, respectively. Overall, this study portrays a basin that will experience increased drought and food scarcity in the future. This information can be used to inform the development of adaptation strategies to improve food and disaster management, create vulnerability maps, establish food monitoring systems in the wake of drying conditions, establish early warning systems, build resilient infrastructure, and address other socio–economic impacts to transition from a vulnerable to a resilient future.

**Author Contributions:** Conceptualization, S.K., B.-S.Z. and O.S.; Methodology, S.K., B.-S.Z. and O.S.; Data curation, R.M.T.; Writing—original draft, S.K.; Writing—review & editing, S.K., O.S., D.P., N.N. and R.M.T. All authors have read and agreed to the published version of the manuscript.

**Funding:** This research received no external funding.

**Informed Consent Statement:** Not applicable.

**Data Availability Statement:** No new data were created or analyzed in this study. Data sharing is not applicable to this article.

**Conflicts of Interest:** The authors declare no conflict of interest.

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
