# Peer review of "Impacts of Climate Change on Hydrological Regimes in the Congo River Basin"

_sustainability, doi:10.3390/su15076066_

Round 1

Reviewer 1 Report

Minor observations

Write the references complete, For example at reference [24] the journal for that paper is Journal of Hydrology. Add the journals to all the papers that are missing.

Line 180. After an equation “Where” is written with small letter and aligned left. Do this for all the equations.

            All variables must be written italic, in the text as in the equations.

In eq. 1 is Qgw, in explanation is Qwg.

Lines 382, 396, 404. Add after Karam et al. (2022) the number of the reference, “[28]”.

Author Response

Thank you very much for your valued comments, they have helped to improve the quality of the paper. A detailed response to your comments has been attached as a pdf file. Please see the attachment.

Best regards,

Sara K.

Reviewer 2 Report

Current climate change on our planet, expressed in global warming, has a diverse impact on the Earth's climate system. Climate change manifests itself in various ways in different parts of the world. In particular, global warming is affecting the planetary hydrological cycle, redistributing the global precipitation field. In the tropics, studies show, the rainfall field heterogeneity is expected to increase, i.e. dry areas and seasons will be drier while wet territories and seasons will be more humid. A peer-reviewed manuscript examining the impact of climate change on hydrological regime in the Congo Basin from the present to 2100 is important and relevant. For the analysis, the authors used reanalysis data and the results of numerical modelling (climate change projections for RCP4.5 and RCP8.5 scenarios) obtained via global and regional climate models.

 I have a few questions regarding the methodology used in the article to analyse the data.

First, it is not clear from the text how the sensitivity analysis was performed. The procedure should be described in more detail. Secondly, the choice of the set of parameters with respect to which the sensitivity analysis was performed requires some justification. The authors limited themselves to referring to previously published articles. In my opinion, that's not enough. The spatial resolution of regional climate models used in the study is quite large (about 50 km). Even modern global climate models have smaller space resolution. Some comments are required.

 Results obtained are quite interesting and will be helpful for planning socio-economic development in the region.

Text needs to be edited. For example, equation (1) should be followed by a comma then followed by a lowercase letter (“where SW is soil…”). Equation (2) should be followed by a dot. Similar typos should be fixed across the text.

Author Response

(The authors gave the same response as above.)

Reviewer 3 Report

Line 44 what is meant by 3.7 M km2 explain (correction needed in area units).

Overall, this study is good for future planning to improve food security and avoid major disaster and desiccation periods.

Author Response

(The authors gave the same response as above.)

Reviewer 4 Report

Dear Authors,

Please, see my comments and suggestions below:

·         Please, write a whole paragraph on general and specific objectives.

·         In Materials and Methods, I could not find the stream gauge used for calibration and validation.

·         In Materials and Methods, I could not find how you intend to compare the past hydrological variability (or regime) with the future ones.

·         You did not assess the RCM outputs regarding the past climate in the Basin. Is it not necessary any bias correction?

·         Discussion on model parameter sensitivity analysis is missing.

·         Normally, 70% of the hydrological data are used for calibration and 30% of them for validation in hydrological studies. You use few data for validation, although the length of the discharge time series is about 30 years.

·         You did not comment, but the general trend of the discharge time series was well simulated. However, at the end of the 1990s, surprisingly, the model underestimated the discharge a lot. Why?   

·         Figures 5 and 7 are unnecessary.

·         Periods 1, 2 and 3 are averages over the ensembles of RCMs? It is not clear.

·         Prior to present the impacts on the discharges, you should show what happened to precipitation, temperature, and evapotranspiration in Periods 1, 2 and 3, which will help you to explain better your findings of discharge projections.

·         You should compare your study with previous studies conducted on other large rivers, including those from other climates. It is fundamental to place your findings in the context of research on the impact of climate changes on large river basins.

Author Response

(The authors gave the same response as above.)

Round 2

Reviewer 2 Report

I have no comments anymore.

Author Response

Thank you very much for your review.

Regards, 

Sara Karam

Reviewer 4 Report

The manuscript can be accepted in present form.

Author Response

Thank you very much for your review.

Best regards,

Sara Karam